# Differential expression of *groEL-1*, *incB*, *pyk-F*, *tal*, *hctA* and *omcB* genes during *Chlamydia trachomatis* developmental cycle

**Gugulethu F. Mzobe**[1,2]*, **Sinaye Ngcapu**[1,2], **Bronwyn C. Joubert**[1], **Willem A. Sturm**[1]

**1** Centre for the AIDS Programme of Research in South Africa (CAPRISA), Durban, South Africa,
**2** Department of Medical Microbiology, School of Laboratory Medicine and Medical Science, University of KwaZulu-Natal, Durban, South Africa

* gugulethu.mzobe@caprisa.org

**Data Availability Statement:** All relevant data are within the manuscript and its Supporting Information files.

## Abstract

*Chlamydia trachomatis* infects squamous and columnar epithelia at the mucosal surface. Research on gene expression patterns of *C. trachomatis* has predominantly focused on non-native host cells, with limited data on growth kinetics and gene expression of chlamydia in keratinocytes. Here, we investigated whether early, mid, and late chlamydial genes observed in HeLa cell line studies were co-ordinately regulated at the transcriptional level even in the keratinized cell line model and whether the expression was stage-specific during the developmental cycle. HaCaT cell lines were infected with chlamydia clinical isolates (US151 and serovar E) and reference strain (L2 434). Expression of *groEL-1*, *incB*, *pyk-F*, *tal*, *hctA*, and *omcB* genes was conducted with comparative real-time PCR and transcriptional events during the chlamydial developmental cycle using transmission electron microscopy. The relative expression level of each gene and fold difference were calculated using the $2^{-\Delta\Delta CT}$ method. The expression of *groEL-1* and *pyk-F* genes was highest at 2 hours post-infection (hpi) in the L2 434 and serovar E. The expression of *incB* gene increased at 2 hpi in L2 434 and serovar E but peaked at 12 hpi in serovar E. L2 434 and US151 had similar *tal* expression profiles. Increased expression of *hctA* and *omcB* genes were found at 2 and 36 hpi in L2 434. Both clinical isolates and reference strains presented the normal chlamydial replication cycle comprising elementary bodies and reticulate bodies within 36 hpi. We show different gene expression patterns between clinical isolates and reference strain during *in vitro* infection of keratinocytes, with reference strain-inducing consistent expression of genes. These findings confirm that keratinocytes are appropriate cell lines to interrogate cell differentiation, growth kinetics, and gene expression of *C. trachomatis* infection. Furthermore, more studies with different clinical isolates and genes are needed to better understand the Chlamydial pathogenesis in keratinocytes.

**Funding:** This study was funded by grants from the National Research Foundation (SFH14061869970) and The University of Kwa-Zulu Natal College of Health Sciences. GFM is funded with a postdoctoral fellowship from the National Research Foundation (SFP180507326699). The funders had no role in study design, data collection and analysis, decision to publish, or preparation of the manuscript.

**Competing interests:** The authors have declared that no competing interests exist.

## Introduction

In 2018, more than a million *Chlamydia trachomatis* infections a day were reported to the World Health Organization, making it a major public health concern in both developed and developing countries [1]. *Chlamydia spp.*, commonly infect a more vulnerable single layer of simple columnar epithelium or non-keratinized stratified squamous epithelium found in the endocervix and transformation zone of the female genital tract [2, 3]. The *C. trachomatis* infection of the genital tract is characterized by sexually transmitted diseases such as oculogenital and lymphogranuloma venereum (LGV) [4, 5]. The ability of this pathogenic bacterium to cause disease is related to its unique two-phase developmental cycle, which takes place in the host cell. In the extracellular phase, the organism manifests as the elementary body (EB), an extracellular and metabolically inert form of *C. trachomatis* that targets columnar or squamous cells lining the mucosal epithelium by receptor-mediated endocytosis [6]. Once the EB is endocytosed it differentiates into the metabolically active reticulate body (RB) which replicates within inclusions by binary fission [6–9]. After many rounds of binary fusion, the majority of RBs convert to the metabolically inactive EBs that are released from the host cell to infect neighboring cells [6–8].

Several studies have shown that multiple chlamydial genes are temporally expressed and regulated by specific mechanisms during the developmental cycle [10–12]. *De novo* transcription and translation are required to facilitate and coordinate molecular events occurring during the developmental cycle [6]. *In vitro* analysis suggest that the *C. trachomatis* developmental cycle is regulated at the transcriptional level in three temporal phases, with early classes of genes expressed within 2 hours post endocytosis, mid-cycle genes within 6–24 hours post-infection (hpi) and during RB replication, and late genes within 24–48 hpi when a majority of RBs convert to EBs [10, 11]. For example, *incB* and *groEL* are genes expressed at high levels in the early stages of the chlamydial infection and continue to be expressed in a low level throughout the developmental cycle [11, 13]. This regulated expression of early genes suggests that they play an important function in the initial stages of infection and not necessarily throughout the developmental cycle. Mid-cycle genes [*ompA*, pyruvate kinase (*pyk*) and trans-aldolase (*tal*)], associated with energy metabolism, are crucial immunogenic determinants for different serovars and subspecies [10, 14–16]. Late genes such as *omcA*, *omcB*, *hctA*, and *hctB* have functions associated with the morphologically dramatic events that occur 24 hpi when RBs convert to EBs. This is confirmed by two abundant cysteine-rich proteins that form part of the outer membrane of EBs and not RBs [17]. In addition, late genes *hctA* and *hctB* encode lysine-rich proteins with a primary sequence similar to the eukaryotic histone Hc1 and Hc2, which mediate the compression of DNA observed when RBs convert into EBs [16, 18–20]. Taken together, this suggests that the temporal expression profile of chlamydial genes demonstrates that genes are transcribed only at a time in the developmental cycle when they are needed.

Although research on pathogenesis and transcriptional expression of genes of *C. trachomatis* is well characterized, it is predominantly conducted in monolayer cultures of epithelial cells such as cervical cancer HeLa cells, which are different from the lower mucosal epithelium infected *in vivo* [10, 21, 22]. There is a growing body of evidence that keratinocytes, a primary target of infection for *C. trachomatis* LGV biovar, can support the growth of *C. trachomatis* in the *in vitro*-generated monolayers of immortalized human keratinocyte (HaCaT) cells [2, 4, 23]. Previously, our group has shown that *C. trachomatis* LGV biovar L2 grew significantly faster than LGV biovar L1 and L3 in HaCaT cells [23]. Nogueira *et al.* (2017) also observed the optimal growth of *C. trachomatis* seeded in the stratified squamous epithelium [2]. Furthermore, another study assessing the effects of *Chlamydia* type III effector TarP on epithelium

was able to grow *C. trachomatis* in various cell types including HaCaT cells [24]. HaCaT cells are one of the cell lines than can be used to interrogate cell differentiation, growth kinetics, and gene expression of *C. trachomatis* infection.

While keratinocytes are optimal cells for chlamydial growth, none of the temporally regulated genes expressed during the chlamydial developmental cycles were investigated in studies using non-transformed keratinocyte epithelial cells, which are important native host cells to gain insights into the *C. trachomatis* cell biology and pathogenesis during infection of squamous epithelium. In this study, we tested the hypothesis that the three temporal classes of early, mid, and late chlamydial genes observed in HeLa cell line studies are co-ordinately regulated at the transcriptional level even in the keratinized cell model. In addition, we determined whether expression of *groEL-1*, *incB*, *pyk-F*, *tal*, *hctA* and *omcB* genes during *in vitro* infection of the HaCaT cells were stage-specific during the developmental cycle.

## Materials and methods

### Cell line and *Chlamydia trachomatis* strains

The HaCaT cell lines (kindly donated in 1995 by Professor N. E. Fusenig of the Cancer Research Centre, Heidelburg, Germany) were used for both propagation of chlamydia and the experimental work. Two *C. trachomatis* LGV and one genital strain were used for the experiments: the L2 reference strain 434 (ATCC® VR-902BTM), one serovar L2 clinical isolate (US151), and one serovar E clinical isolate, respectively. US151 and serovar E were isolated in the male patient presenting with genital ulcer and urethritis at the Prince Cyril Zulu Communicable Diseases Clinic in Durban, South Africa.

### *C. trachomatis* culture and infectious particle recovery

*C. trachomatis* strains were propagated in HaCaT cell monolayers. Briefly, HaCaT cells grown in 12-well plates ($2x10^6$ cells per well) were infected with *C. trachomatis* in RPMI-1640 supplemented with glucose (5.4 mg/ml), 10% FBS, 10 mM HEPES, 2mM L-glutamine, gentamicin (10 μg / ml) and amphotericin-B (5 μg / ml). Cultures were centrifuged for 1 hour and then incubated for another hour at 37˚C, 5% $CO_2$. Media change was performed after the 1-hour incubation, then cultures containing fresh medium were incubated at 37˚C with 5% $CO_2$ for 48 hours. After 48 hours of growth, infected host-cell monolayers were lysed and chlamydia was harvested in sucrose phosphate glutamate (SPG) buffer. Triplicates of infected HaCaT cell monolayers cultured with tenfold serial dilutions of the inoculum were used to determine the infectious titer. *C. trachomatis* LGV L2 434, L2 US151, and strain E were used to infect the HaCaT monolayers at a multiplicity infection (MOI) of 10 and incubated at 37˚C. The number of chlamydial inclusions was enumerated, and the number of inclusion-forming units/ml was calculated. Infection was confirmed using the MicroTrak *C. trachomatis* Culture Confirmation Test kit (Trinity Biotech) and fluorescence microscopy.

### Isolation and amplification of *C. trachomatis* genes

Infected HaCaT cells lysed in 2 ml/well of guanidine thiocyanate (GTC) solution containing 1.4% β-mercaptoethanol (Sigma, Steinheim, Germany) and 200 μl Trisure (Bioline, London, UK) were used to isolate total *C. trachomatis* RNA harvested at 2, 12, 24, 36 and 48 hpi. Complementary DNA was generated using adjusted RNA (0.2 μg/μl) and the high capacity cDNA reverse transcription kit (Applied Biosystems, Life Technologies. *C. trachomatis* genes (*groEl-1*, *incB*, *pyk-F*, *tal*, *omcB* and *hctA*) were amplified by comparative real-time PCR, using primers (Table 1), PCR conditions and ran on ABI Prism 7500 Real-Time PCR System (Applied

**Table 1. Primer sequences of the *groEl-1*, *incB*, *pyk-F*, *tal*, *omcB*, *hctA* genes and 16S rRNA reference gene used in the real time PCR.**

| Primer | Forward (5' - 3') | Reverse (5' - 3') |
|---|---|---|
| *16S rRNA* | TCGAGAATCTTTCGCAATGGAC | CGCCCTTTACGCCCAATAAA |
| *incB* | CCCCTCGAGGGATGGTTCATTCTGTATACAATTCATTG | CCCGAATTCCTATTCTTGAGGTTTTGTTGGGCTG |
| *groEL-1* | CGGGGTACCTTAAGGAGCGCATCAATGG | CGGGGTACCGGCTCGAAGAATCTATTTGTTCC |
| *pyk-F* | ATCGCTGCTTGTTCGTAGATGTAATG | CCCTTATGTTAGAGAACGAGCTAATG |
| *tal* | GCAGCGATCCACCAATCATAAATCCGACA | CCGAAATACGCTCTCTGCAACTCCACA |
| *hctA* | ACCGAATTCTTTTCTATTAACAGAGGAAAAATAACCTA | TTTAATTTTTAATTAGTTTGTTTGTTCAAA |
| *omcB* | GTGATGGGAAATTAGTCTGG | ATCCTGTGTTCACTACTTCG |

Biosystems, Life Technologies), as previously described by Goldschmidt et al., 2006. The 16S rRNA gene was used as a reference gene.

## Transmission Electron Microscopy (TEM)

*C. trachomatis* infected HaCaT cells grown on 12 mm Nunc™ Thermanox™ coverslips (ThermoFisher, USA) were fixed with 2% glutaraldehyde in EMEM (BioWhittakerTM, Walkersville, USA), followed by two washing steps with EMEM and one step with 0.1M sodium cacodylate buffer, pH 7.4. The cells were fixed with 1% osmium tetraoxide, dehydrated with a graded series of ethanol (50, 70, 90, and 100%) at 24˚C, infiltrated, and embedded in Spurr's resin (Sigma, Steinheim, Germany) overnight at 60˚C (S1 Table). A beam capsule filled with Spurr resin was used to embed the fixed cells for TEM. Ultrathin sections (50–60 nm) were cut and collected onto 3.05 mm diameter, square uncoated mesh copper TEM grids, then double-stained with uranyl acetate and Reynold's lead citrate (Sigma, Steinheim, Germany) for 10 minutes each [25]. Sections were viewed using a Jeol 1010 transmission electron microscope (JEOL Ltd) at an accelerating voltage of 100 kV. The TEM was interfaced with a Megaview III Software Imaging Systems camera unit (Soft Imaging System, Münster, Germany). Images were captured digitally, and measurements performed using iTEM analySIS (Soft Imaging System, Münster, Germany) image analyzing software.

## Data analysis and validation analysis of standard curves for *C. trachomatis* genes

All experiments were carried out three times in triplicates. The $2^{-\Delta\Delta CT}$ method was used to analyze the relative expression level of each gene from the real-time quantitative PCR experiments. Briefly, the mean $\Delta CT$ values were calculated by subtracting the mean target $C_T$ value from the mean 16S rRNA reference $C_T$ value using the formula: $\Delta CT$ ($C_{Ttargetgene} - C_{T16SrRNA}$) [26]. The $\Delta C_T$ values were plotted versus log input amount cDNA to create a semi-log regression line. The absolute slope value of the semi-log regression line close to zero was used as a general criterion for passing a validation experiment. The standard variance of the $\Delta C_T$ was calculated from the standard deviations of the target gene and reference values using the formula: $s = (s_1^2 + s_2^2)^{1/2}$, where the square root of X is $X^{1/2}$ and s, the standard deviation. In addition, $\Delta\Delta C_T$ values were calculated by subtracting $\Delta C_T$ value of the test samples (L2 434, US151, and serovar E) from the calibrator sample (L2 reference strain) using the formula: $\Delta\Delta C_T = \Delta C_T$ test sample - $\Delta C_T$ calibrator sample. Furthermore, the standard deviations of the $\Delta\Delta C_T$ values were the same as the standard deviation of the $\Delta C_T$ value. This study did not formally test any hypothesis and therefore could not produce any p-values due to limited data points. However, we adopted a simple analytical approach by incorporating standard deviations of the $\Delta\Delta C_T$ values into the fold difference calculation ($\Delta\Delta C_T \pm s$) [26] and calculated the relative

difference between target and reference values. All analyses were conducted using Microsoft Excel (Microsoft, USA) and GraphPad Prism 8 (GraphPad Software, USA).

## Results

### Differential expression of *C. trachomatis* genes at 37°C post-infection by serovars

Here, we determined whether HaCaT gene expression patterns differed between clinical isolates and reference strain and whether the expressions were time-dependent. Overall, we observed differences in the gene expression patterns between clinical isolates and reference strain (Fig 1 and S2 Table). There were continuous expression of *groEL-1*, *incB*, and *pyk-F* genes while *hctA*, *tal*, and *omcB* genes were expressed at low levels following infection of HaCaT cells with all strains. Early- (*groEL-1*) and mid-cycle (*pyk-F*) genes were expressed between 2 and 48 hpi and the expression peaked by more than 90 fold at 2 hpi with L2 434 and

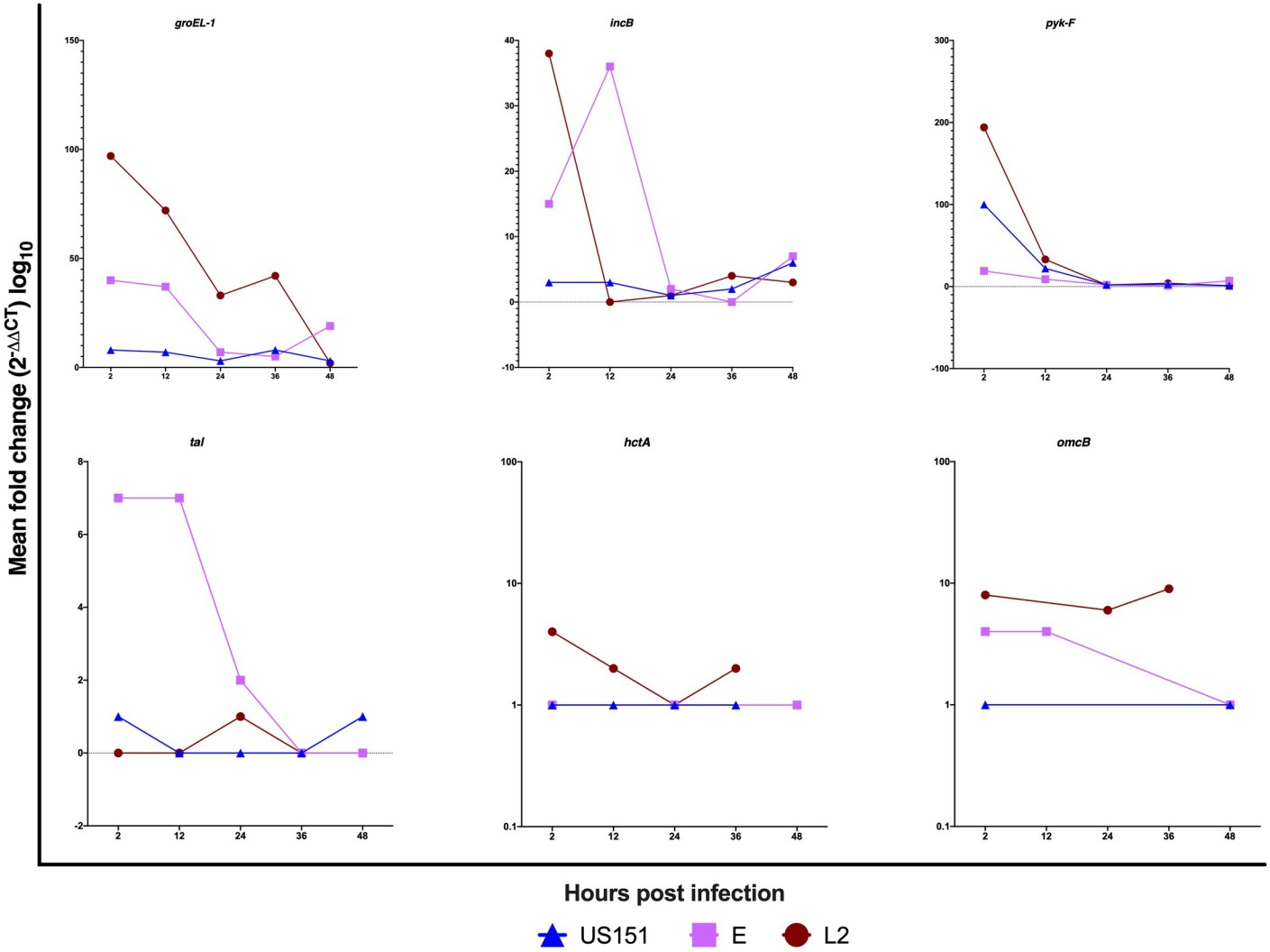

**Fig 1. Line graphs representing the mean fold difference ($2^{-\Delta\Delta CT}$) of the *groEL-1*, *incB*, *pyk-F*, *hctA*, *tal* and *omcB* genes of *C. trachomatis* post infection in HaCaT cell lines with reference strain L2 434 and clinical isolates US151 (L2) and serovar E at 2, 12, 24, 36 and 48-hours.** The standard deviations of the $\Delta\Delta C_T$ values were incorporated into the fold difference calculation and the results are based on experiments carried out three times in triplicates.

US151 (L2). In addition, *groEL-1* and *pyk-F* genes were moderately expressed in serovar E. The expression of the early-cycle gene *incB* was high with more than 30 fold at 2 hpi in L2 434 and and at 12 hpi in US151 (L2). There was a less than 7 fold expression of *tal* and 4 fold expression of *omcB* gene at 2 and 12 hpi in serovar E stimulated cells. Although modest, the expression of late-cycle genes *hctA* and *omcB* were observed at 2 hpi and the expression of *omcB* gene peaked at 36 hpi in L2 434. Low expression levels of *tal*, *hctA* and *omcB* were observed in US151 (L2). Similarly, L2 434 did not induce high levels of *tal* gene. A similar expression profile (4 fold) of *omcB* was observed between 2 and 12 hpi in cells infected with serovar E. Some of the genes in response to serovar infection could not be expressed at 2 (strain L2: *tal*), 12 (strain L2: *inc B*, *tal*, *omcB*; serovar E: *hctA*; US151: *tal*, *omcB*), 24 (serovar E: *omcB*; US151: *tal*, *omcB*), 36 (serovar E: *incB*, *pyk-F*, *tal*, *hctA*, *omcB*; US151: *tal*, *omcB*), 48 hours post-infection (strain L2: *tal*, *hctA*, *omcB*; serovar E: *tal*; US151: *htcA*) using qPCR.

### Expression of *C. trachomatis groEL-1*, *pyk-F* and *hctA* genes in the HeLa cells

We further verified the observations in HaCaT cells using HeLa cells. Fig 2 summarizes the results of RT-PCR analysis in infected HeLa cells for three chlamydial genes (*groEL-1*, *pyk-F*, *hctA*) representing each of the three proposed temporal classes of chlamydial gene expression. The findings of this study were in keeping with the published data [10], with *groEL-1*, *pyk-F* and *hctA* detected between 2 and 48 hpi at 37°C in both cell lines. HaCaT cells expressed high levels of *groEL-1* and *pyk-F* genes at 2 and 24 hpi compared to HeLa cells. Expression of *hctA* gene at 24 hpi was more pronounced in HeLa compared to HaCaT cells and continued to 48hpi with L2 434 and US151 (L2). Some of the genes in response to serovar infection could not be expressed at 2 (strain L2: *pyk-F*, *hctA*; US151: *pyk-F*, *htcA*) and 12 hours (L2: *hctA*; US151: *hctA*) post infection using qPCR, which is in keeping with published data.

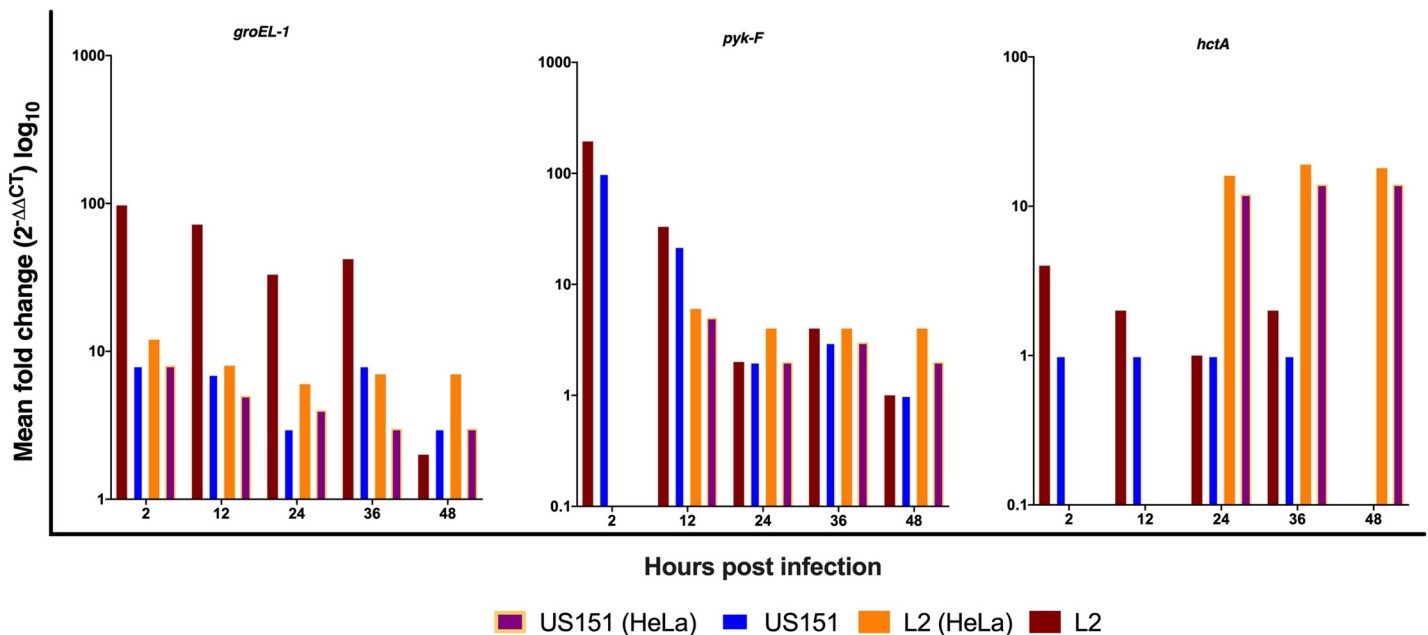

**Fig 2. The expression *groEL-1*, *pyk-F* and *hctA* genes of *C. trachomatis* reference strain L2 434 and clinical isolates US151 (L2) and serovar E at 2, 12, 24, 36 and 48 hpi in HaCaT and HeLa cells at 37°C.** Bar charts representing the mean fold difference (2 $^{-\Delta\Delta CT}$) of genes and the results are based on experiments carried out three times in triplicates.

## The developmental cycle of *C. trachomatis* in human keratinocytes

To place transcriptional events of *groEL-1*, *incB*, *pyk-F*, *hctA*, *tal* and *omcB* genes into the context of the *C. trachomatis* LGV and genital biovars developmental cycle, we conducted an ultrastructural analysis of the chlamydial developmental cycle in HaCaT cells using TEM. Both LGV (L2 434 and US151) and the genital biovar (serovar E) presented the normal chlamydial replication cycle comprising EB and RB. Fig 3 shows the ultrastructural events observed during *C. trachomatis* LGV and genital biovars developmental cycle at 37°C. Fig 3 shows the ultrastructural events of strain L2 434, US151 (L2), and serovar E. Within Fig 3, panel A1, B1, and C1 show uninfected polygonal-shaped HaCaT cells with a nucleus containing several nucleoli. Infected HaCaT cells in panel A2 appeared similar to uninfected cells at 2 hpi with no noticeable morphological differences. Strain L2 434 and US151 (L2) replicated faster serovar E, with approximately 26 matured RBs dividing by binary fission in the inclusion within 12 hpi (A3, B3, and C3). EBs appeared within 36 hpi (A5). EBs continued to accumulate within the inclusion at 48 hpi (A6). Similar ultrastructural events to L2 434 were observed in HaCaT cells infected US151 (L2) (Fig 3) from 2 to 24 hpi. However, there were numerous EBs at 36 and 48 hpi observed in HaCaT cells infected with L2 434 (panel A5 and A6) compared to US151 (L2) (B5 and B6) and serovar E (C5 and C6).

## Discussion

The chlamydial developmental cycle has been well characterized microscopically. However, the signals that stimulate conversion from EB to RB and mechanisms associated with the regulation of intracellular development remain unclear [27]. It has been shown that the synthesis of numerous proteins occurs throughout the chlamydial developmental cycle, while other proteins are strictly associated with mid- and late-stage differentiation [10]. Early-, mid- and late-cycle genes represent a subset of genes that are important in understanding key events in the differentiation processes that control the developmental cycle [12]. Here, we used native host cell lines to investigate the *in vitro* chlamydial gene expression during the developmental cycle to understand the different pathogenicity of the LGV biovar of *C. trachomatis* and its G(genital) biovar in keratinocytes 37°C. This study used HaCaT cell line, ATCC reference strain (L2 434), and 2 clinical isolates (LGV L2 US151 and genital biovar E) isolated in a male patient presenting with genital ulcer and urethritis [23]. We observed different transcriptional expression of chlamydial genes post-infection with L2 434, US151, and serovar E. In addition, L2 434, US151, and serovar E strains presented the normal chlamydial replication cycle comprising EB and RB within 36 hpi.

For the first time, the *in vitro* expression of the *groEL-1*, *incB*, *pyk-F*, *hctA*, *tal* and *omcB* genes of *C. trachomatis* reference strain L2 434 and clinical isolates US151 (L2) and serovar E were studied using HaCaT cells. *groEL-1* expression levels remained fairly constant between 2 and 36 hpi for all tested strains. However, strain E showed increased *groEL-1* expression levels at 48 hpi. g*roEL-1* is known to increase during nutrient deprivation which occurs early in the developmental cycle [13]. The increase observed in *groEL-1* at 48 hpi may be attributed to the loss of energy and nutrients during the differentiation of RBs to EBS late in the developmental cycle [13]. Cells infected with L2 434 had over a 30-fold increase in *incB* expression level at 2 hpi, compared to L2 US151 in which a constantly lower level of expression was observed. This was in agreement with the chlamydial growth rate observed using the TEM. Lack of induced expression of *incB* in our clinical LGV isolate may suggest that this isolate use one or more *inc* proteins other than *incB*. In addition, the difference between reference strain and clinical isolates may due to the modification of strain LGV 434 during numerous passages in cell culture. Furthermore, *incB* expression levels peaked at 12 hpi for strain E. The difference in the pattern

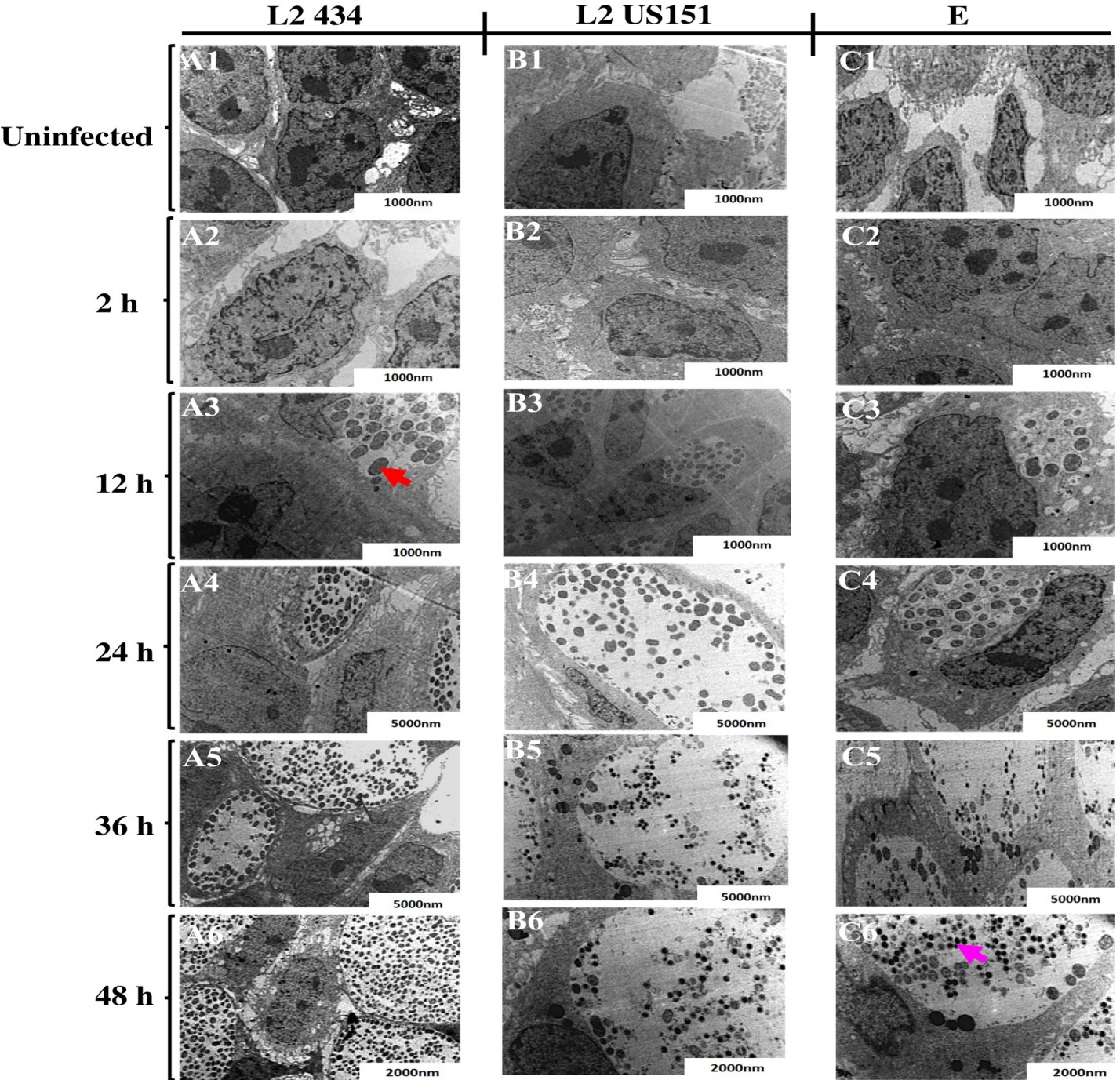

**Fig 3. TEM micrographs of HaCaT cells infected with *C. trachomatis* reference strain L2 434, clinical isolates US151, and serovar E demonstrating differentiation, growth, division and redifferentiation at 37°C over the course of the developmental cycle.** Panel A1, B1 and C1 show micrographs of uninfected HaCaT cells (negative controls), A2, B2 and C2 at 2 hpi, A3, B3 and C3 at 12 hpi, A4, B4 and C4 at 24 hpi, A5, B5 and C5 at 36 hpi, A6, B6 and C6 at 48 hpi. Cultures were infected at an MOI of 10. Red arrowhead point at RB and pink arrowhead point at EB compartmentalized within the inclusions.

of gene expression observed for strain E could be correlated with the fact that the primary target for genital serovars are epithelial cells of the genital tract, and in this study, keratinocytes were used.

The expression of *tal* and *pyk-F* genes was observed at different levels amongst the tested chlamydial strains. *pyk-F* expression was most abundant at 2 hpi in all tested chlamydial strains, indicating that energy is required immediately following the invasion of chlamydia into the keratinocytes. In contrast, there was no expression of *pyk-F* in the early phase in HeLa cells infected with serovar E. The expression of *tal* was highest at 2 and 12 hpi in HaCaT cells with strain E, which correlated with rapid growth and division of RBs as observed in the ultra-structural analysis of the chlamydial developmental cycle. This was also consistent with the TEM observations. Our results suggest that *tal* and *pyk-F* genes were required to support energy metabolism and growth throughout the chlamydial developmental cycle in keratinocytes.

Two late-cycle genes that have been reported to be expressed from 24 hpi were also analyzed. These are, and *omcB* that encodes a cysteine-rich outer membrane protein that interacts with the MOMP to form this complex which involves extensive protein cross-linking through the formation of cysteine bonds [12]. We also observed high expression of *hctA*, a gene that encodes a chlamydial histone-like protein and mediates chromosomal condensation during the differentiation of RBs to EBs, at 2 hpi and decreased by 1- fold at 12 hpi for L2 434 and serovar E. High *hctA* expression observed in this study could be due to the highly condensed chlamydial chromosome in the early phase of the developmental cycle. Previously, it has been shown that the expressions of *hctA* gene was reduced as condensed chromatin of EB was dispersed during differentiation into pleomorphic RB [20]. Previous studies reported that *hctA* is not expressed until 24 hpi in HeLa cells [10, 11]. However, our study shows that *hctA* was expressed throughout the chlamydial developmental cycle but the level of expression increases from 24 hpi in keratinocytes. Another gene (*omcB*) expressed in the late-cycle was also observed at 2 hpi with L2 434. At 12 hpi the expression level was suppressed but upregulated again at 24 hpi. The expression of *omcB* gene is only found in EBs as the component of the disulfide-linked outer membrane protein complex that confers structural stability to EBs [10]. Thus, the early expression of *omcB* observed in this study suggests that chlamydia was still in an EB form at 2 hpi in keratinocytes. This was consistent with the TEM, which showed numerous EBs at 36 hpi.

Taken together, the observed differences in transcriptional expression of chlamydial genes between reference strain (L2 434) and clinical isolates (US151 and serovar E) and inconsistency with published data may not be attributed exclusively to the type of model or the MOI used, but to several other factors. These include carryover mRNA that may be present in EBs, high-passage numbers, different regulatory systems for gene expression in HaCaT cells compared to HeLa cell lines, and the origin (site of infection) of the isolated strain. Furthermore, in our population, LGV presents in the primary stage as a painful genital ulcer without the tendency to resolve spontaneously. This could mean that there is a genetic difference between LGV strains that is responsible for this difference in clinical presentation [5]. Lastly, the mRNA decay rate was not investigated in this study.

## Conclusion

Our study showed different gene expression patterns between clinical isolates and reference strain during *in vitro* infection of the immortalized human keratinocyte, suggesting that keratinocytes are also appropriate to interrogate cell differentiation, growth kinetics, and gene expression of *C. trachomatis* infection. Extensive understanding of the chlamydia intracellular biology, including temporal expression patterns of genes during *in vitro* and *in vivo* infections is needed to develop novel therapeutic strategies against *C. trachomatis* infections and disease.

## Supporting information

**S1 Table. Processing schedule for TEM.**
(DOCX)

**S2 Table. Mean fold changes for six chlamydial genes in HaCaT cells.**
(DOCX)

## Acknowledgments

We acknowledge the Department of Microscopy and Microanalysis Unit in the University of KwaZulu Natal, Westville campus, for their significant contribution to the development of TEM micrographs.

## Author Contributions

**Conceptualization:** Gugulethu F. Mzobe, Bronwyn C. Joubert, Willem A. Sturm.

**Data curation:** Gugulethu F. Mzobe, Sinaye Ngcapu, Bronwyn C. Joubert, Willem A. Sturm.

**Formal analysis:** Gugulethu F. Mzobe, Sinaye Ngcapu, Bronwyn C. Joubert, Willem A. Sturm.

**Funding acquisition:** Gugulethu F. Mzobe, Bronwyn C. Joubert, Willem A. Sturm.

**Investigation:** Gugulethu F. Mzobe, Bronwyn C. Joubert, Willem A. Sturm.

**Methodology:** Gugulethu F. Mzobe, Bronwyn C. Joubert, Willem A. Sturm.

**Project administration:** Gugulethu F. Mzobe, Bronwyn C. Joubert, Willem A. Sturm.

**Resources:** Bronwyn C. Joubert, Willem A. Sturm.

**Supervision:** Bronwyn C. Joubert, Willem A. Sturm.

**Validation:** Bronwyn C. Joubert, Willem A. Sturm.

**Visualization:** Gugulethu F. Mzobe, Bronwyn C. Joubert, Willem A. Sturm.

**Writing – original draft:** Gugulethu F. Mzobe, Sinaye Ngcapu, Bronwyn C. Joubert, Willem A. Sturm.

**Writing – review & editing:** Gugulethu F. Mzobe, Sinaye Ngcapu, Willem A. Sturm.

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
