## [Decision Letter · Decision Letter 0]

24 Feb 2021

PONE-D-21-02174

Differential gene expression by Chlamydia trachomatis in keratinocytes at different temperatures

PLOS ONE

Dear Dr. Mzobe,

Thank you for submitting your manuscript to PLOS ONE. After careful consideration, we feel that it has merit but does not fully meet PLOS ONE’s publication criteria as it currently stands. Therefore, we invite you to submit a revised version of the manuscript that addresses the points raised during the review process.

We look forward to receiving your revised manuscript.

Kind regards,

Michael F Minnick, PhD

Academic Editor

PLOS ONE

Journal Requirements:

'..This study was funded by grants from the National Research Foundation (SFH14061869970) and The University of Kwa-Zulu Natal College of Health Sciences. GFM is funded with a postdoctoral fellowship from the National Research Foundation (SFP180507326699).'

'Author: G.F Mzobe

Funder: National Research Foundation (www.nrf.ac.za)

Grant number: SFH14061869970

The funders had no role in study design, data collection and analysis, decision to publish, or preparation of the manuscript.'

5. Please include your table 1 as part of your main manuscript and remove the individual file. Please note that supplementary tables should remain as separate "supporting information" files.

6. Please include captions for your Supporting Information files at the end of your manuscript, and update any in-text citations to match accordingly. Please see our Supporting Information guidelines for more information: http://journals.plos.org/plosone/s/supporting-information

Additional Editor Comments:

While there is certainly value in examining alternate host cells and various serovars of Chlamydia for analysis of expression patterns, the manuscript contains a number of major issues that must be addressed before it is suitable for publication.

The two reviewers have disparate evaluations of your manuscript and both are experts in the field. Unfortunately, my analysis of the paper falls more in line with reviewer 1. My major concerns are as follows: 1) There are missing data in both figures without any explanations, 2) lack of statistical analyses for comparing data, and 3) inconsistencies between the data in your study and to previously-published results on temporal gene expression in Chlamydia (see citation [10]). For example, the late-cycle genes examined (hctA, omcB) are only modestly expressed at all the time points, and a mid-cycle gene (pykF) peaked at 2 hpi.

Reviewers' comments:

Reviewer's Responses to Questions

**Comments to the Author**

1. Is the manuscript technically sound, and do the data support the conclusions?

Reviewer #1: No

Reviewer #2: Yes

2. Has the statistical analysis been performed appropriately and rigorously? 

Reviewer #1: No

Reviewer #2: Yes

3. Have the authors made all data underlying the findings in their manuscript fully available?

Reviewer #1: Yes

Reviewer #2: Yes

4. Is the manuscript presented in an intelligible fashion and written in standard English?

Reviewer #1: No

Reviewer #2: Yes

5. Review Comments to the Author

Reviewer #1: This manuscript examines developmental gene regulation of 4 Chlamydia trachomatis strains in an immortalized human keratinocyte cell line. Six chlamydial genes shown in multiple studies to be differentially expressed throughout the developmental cycle are used to analyze temporal gene expression at 5 different time points. There may be some utility in this study for those studying chlamydial strains infected keratinocytes. However, the results really don't follow any patterns established in the literature and do not appear to be consistent between Figs 1 and 2. I have only a few comments for consideration.

1. line 87. The HaCaT cell line may be useful in specific instances but I doubt that it is "the most appropriate cell line" to use. Statements of significance or priority should be minimized.

2. There are no statistics for Figures 1 or 2. The legend of Fig 1 states that the standard deviations were incorporated into the fold-difference calculations. Perhaps a more qualified statistician might be needed to evaluate this but the statement on line 163 that " could not produce any p-values due to limited data points"

3. There are missing data points in both figures with no explanation. There are no discernable patterns in Fig. 1. Indeed, in the bottom 3 panels some genes did not change at all. The differences do not really seem congruent with Fig. 2.

4. Caution should be exercised in interpreting results from the earliest time points. Carryover mRNA is known to be present in EBs and not due to early transcription.

5. line 200. "G (E) strain" What is this? Does G represent genital. If so, please spell it out and identify serovars as such.

6. Urogenital strains are generally slower growing than LGV strains. Some of the LGV isolates seem to have different patterns of expression. A better way to get at this would be one step growth curves plotting numbers of progeny IFUs over time.

7. Fig 3. Scale bars are unreadable and it appears that different magnifications are shown. It would be easier to evaluate if all images were at a similar magnification.

8. The manuscript would benefit from editing for English grammar.

Reviewer #2: The manuscript by Mzobe and colleagues addresses chlamydial gene expression in keratinocytes. The work is thorough and conducted well. The conclusions, though only modestly incremental, are sufficiently significant to warrant publication. The electron microscopy images will be particularly useful to readers.

I have no significant criticisms of the study, the following are minor changes that should be addressed by the authors:

Line 35: keratinocytes should be plural

Line 45: Use Chlamydia spp., and italicize.

Line 49: “can occur either as…” should be changed.

Line 62: indicate which inc

Line 104: “strains” should be used in place of “serovars”. Same with line 120

Line 198: Why use biovar here instead of serovar?

Figure 2: The color for US151 should be changed. The distinction will not be visible to most readers.

There is some funniness in the references that should be worked out.

6. PLOS authors have the option to publish the peer review history of their article (what does this mean?). If published, this will include your full peer review and any attached files.

Reviewer #1: No

Reviewer #2: No

---

## [Author Response · Author response to Decision Letter 0]

9 Mar 2021

Reviewer #1:

This manuscript examines developmental gene regulation of 4 Chlamydia trachomatis strains in an immortalized human keratinocyte cell line. Six chlamydial genes shown in multiple studies to be differentially expressed throughout the developmental cycle are used to analyze temporal gene expression at 5 different time points. There may be some utility in this study for those studying chlamydial strains infected keratinocytes. However, the results really don't follow any patterns established in the literature and do not appear to be consistent between Figs 1 and 2. I have only a few comments for consideration.

1. line 87. The HaCaT cell line may be useful in specific instances but I doubt that it is "the most appropriate cell line" to use. Statements of significance or priority should be minimized.

Response: We have since downplayed the statement and changed to “HaCaT cells are one of the cell lines than can be used to interrogate cell differentiation, growth kinetics, and gene expression of C. trachomatis infection” line 98 of the manuscript

2. There are no statistics for Figures 1 or 2. The legend of Fig 1 states that the standard deviations were incorporated into the fold-difference calculations. Perhaps a more qualified statistician might be needed to evaluate this but the statement on line 163 that " could not produce any p-values due to limited data points"

Response: We have consulted with a qualified senior statistician in our group and she shared the same sentiments that p values can only be produced when there is a hypothesis to be tested. She added that, statistically, p value gives us the probability of obtaining data as extreme as or more extreme than observed given the null hypothesis. On these basis, it mean that the p value and null hypothesis are tied together. Our study is descriptive with limited data points and we don’t intend to produce any inferential statistics.

3. There are missing data points in both figures with no explanation. There are no discernable patterns in Fig. 1. Indeed, in the bottom 3 panels some genes did not change at all. The differences do not really seem congruent with Fig. 2.

Response: We have since provided detailed explanation on missing data points in the text and figure legends in the manuscript. For Fig. 1: Some of the genes in response to serovar infections could not be expressed at 2 (strain L2: tal), 12 (strain L2: inc B, tal, omcB; serovar E: hctA; US151: tal, omcB), 24 (serovar E: omcB; US151: tal, omcB), 36 (serovar E: incB, pyk-F, tal, hctA, omcB; US151: tal, omcB), 48 hours post-infection (strain L2: tal, hctA, omcB; serovar E: tal; US151: htcA) using qPCR. For figure 2: Gene expression profile of certain genes in HeLa cells could not be detect at 2 (strain L2: pyk-F, hctA; US151: pyk-F, htcA) and 12 hours (L2: hctA; US151:hctA) post infection using qPCR, which is in keeping with published data.

Fig. 1 shows gene expression in HaCaT cells only, whereas Fig.2 compares expression of 3 genes (one for each chlamydial developmental stage) in HaCaT (L2: maroon; US151: blue) Vs HeLa (L2: orange; US151: purple) cells. Patterns for gene expression levels in HaCaT cells (Fig. 2) are in agreement with the patterns shown in Fig.1. However, the patterns observed in HeLa cells are in keeping with previously published data and not our HaCaT cell derived findings.

4. Caution should be exercised in interpreting results from the earliest time points. Carryover mRNA is known to be present in EBs and not due to early transcription.

Response: we note the reviewer’s valuable comment and we have added this to our limitation section in the discussion and reads as follows: “Taken together, the observed differences in transcriptional expression of chlamydial genes between reference strain (L2 434) and clinical isolates (US151 and serovar E) and inconsistency with published data may not be attributed exclusively to the type of model or the MOI used, but to several other factors. These include carryover mRNA that may be present in EBs, high-passage numbers, different regulatory systems for gene expression in HaCaT compared to HeLa cell lines, and the origin (site of infection) of the isolated strain.”

5. line 200. "G (E) strain" What is this? Does G represent genital. If so, please spell it out and identify serovars as such.

Response: "G (E) strain" has been changed to “the genital biovar (serovar E)”. This has also been corrected throughout the manuscript. 

6. Urogenital strains are generally slower growing than LGV strains. Some of the LGV isolates seem to have different patterns of expression. A better way to get at this would be one step growth curves plotting numbers of progeny IFUs over time.

Response: this study is an ancillary of a parent study (Joubert and Sturm, 2011) that has done the growth curves plotting numbers of progeny IFUs over time 

Joubert, B. C., and Sturm, A. W. (2011). Differences in Chlamydia trachomatis growth rates in human keratinocytes among lymphogranuloma venereum reference strains and clinical isolates. J. Med. Microbiol. 60, 1565–1569. doi: 10.1099/jmm.0.032169-0

7. Fig 3. Scale bars are unreadable and it appears that different magnifications are shown. It would be easier to evaluate if all images were at a similar magnification.

Response: Scale bars have been made visible. Magnification is similar across each time point.

8. The manuscript would benefit from editing for English grammar.

Response: this has been resolve throughout the manuscript

Reviewer #2:

The manuscript by Mzobe and colleagues addresses chlamydial gene expression in keratinocytes. The work is thorough and conducted well. The conclusions, though only modestly incremental, are sufficiently significant to warrant publication. The electron microscopy images will be particularly useful to readers.

I have no significant criticisms of the study, the following are minor changes that should be addressed by the authors:

1. Line 35: keratinocytes should be plural

Response: “Keratinocyte” has been changed to “keratinocytes”

2. Line 45: Use Chlamydia spp., and italicize.

Response: “Chlamydia” was changed to “Chlamydia spp.”

3. Line 49: “can occur either as…” should be changed.

Response: “can occur either as” to “In the extracellular phase, the organism manifests as the elementary body (EB)”

4. Line 62: indicate which inc

Response: Specified inc protein as incB

5. Line 104: “strains” should be used in place of “serovars”. Same with line 120

Response: “serovars” changed to “strains”

6. Line 198: Why use biovar here instead of serovar?

Response: a combination of serovar can be termed biovar

7. Figure 2: The color for US151 should be changed. The distinction will not be visible to most readers.

Response: The colour for US151 in figure 2 has been changed from red to purple. 

8. There is some funniness in the references that should be worked out.

Response: References have been sorted out

---

## [Decision Letter · Decision Letter 1]

17 Mar 2021

Differential expression of groEL-1, incB, pyk-F, tal, hctA and omcB genes during Chlamydia trachomatis developmental cycle

PONE-D-21-02174R1

Dear Dr. Mzobe,

We’re pleased to inform you that your manuscript has been judged scientifically suitable for publication and will be formally accepted for publication once it meets all outstanding technical requirements.

Kind regards,

Michael F Minnick, PhD

Academic Editor

PLOS ONE

Additional Editor Comments (optional):

Reviewers' comments:

Reviewer's Responses to Questions

**Comments to the Author**

1. If the authors have adequately addressed your comments raised in a previous round of review and you feel that this manuscript is now acceptable for publication, you may indicate that here to bypass the “Comments to the Author” section, enter your conflict of interest statement in the “Confidential to Editor” section, and submit your "Accept" recommendation.

Reviewer #1: All comments have been addressed

Reviewer #2: All comments have been addressed

2. Is the manuscript technically sound, and do the data support the conclusions?

Reviewer #1: Partly

Reviewer #2: (No Response)

3. Has the statistical analysis been performed appropriately and rigorously? 

Reviewer #1: N/A

Reviewer #2: (No Response)

4. Have the authors made all data underlying the findings in their manuscript fully available?

Reviewer #1: Yes

Reviewer #2: (No Response)

5. Is the manuscript presented in an intelligible fashion and written in standard English?

Reviewer #1: Yes

Reviewer #2: (No Response)

6. Review Comments to the Author

Reviewer #1: (No Response)

Reviewer #2: (No Response)

7. PLOS authors have the option to publish the peer review history of their article (what does this mean?). If published, this will include your full peer review and any attached files.

Reviewer #1: No

Reviewer #2: No

---

## [Editor Report · Acceptance letter]

7 Apr 2021

PONE-D-21-02174R1 

Differential expression of *groEL-1, incB, pyk-F, tal, hctA* and *omcB* genes during *Chlamydia trachomatis* developmental cycle 

Dear Dr. Mzobe:

I'm pleased to inform you that your manuscript has been deemed suitable for publication in PLOS ONE. Congratulations! Your manuscript is now with our production department. 

Kind regards, 

on behalf of

Dr. Michael F Minnick 

Academic Editor

PLOS ONE